# Evidence-Based Integrated Intervention in Patients with Schizophrenia: A Pilot Study of Feasibility and Effectiveness in a Real-World Rehabilitation Setting

**DOI:** 10.3390/ijerph17103352

**Published:** 2020-05-12

**Authors:** Gabriele Nibbio, Stefano Barlati, Paolo Cacciani, Paola Corsini, Alessandra Mosca, Anna Ceraso, Giacomo Deste, Antonio Vita

**Affiliations:** 1Department of Clinical and Experimental Sciences, University of Brescia, 25133 Brescia, Italy; gabriele.nibbio@gmail.com (G.N.); ceras.anna@gmail.com (A.C.); antonio.vita@unibs.it (A.V.); 2Department of Mental Health and Addiction Services, ASST Spedali Civili, 25133 Brescia, Italy; cacciani.paolo@tiscali.it (P.C.); corsinipaola13@gmail.com (P.C.); alessanmosca@tiscali.it (A.M.); giacomodeste@mac.com (G.D.)

**Keywords:** schizophrenia, psychosocial rehabilitation, real-world functioning, cognitive remediation, social skills training

## Abstract

Impairment in real-world functioning remains one of the most problematic challenges that people with schizophrenia have to face. Various psychosocial interventions have proven to be effective in promoting recovery and improving functioning in schizophrenia; however, their implementation and their effectiveness in routine rehabilitation practice are still objects of study. The present pilot study aimed to assess the feasibility and effectiveness on clinical and real-world outcomes of an integrated treatment protocol composed of stable pharmacological treatment, computer-assisted cognitive remediation and social skills training provided in a rehabilitation center. Predictors of functional improvement were also assessed. Seventy-two patients diagnosed with schizophrenia participated in the study. A significant (*p* < 0.001) improvement in positive, negative and total symptoms, as well as in global clinical severity and real-world functioning outcomes was observed, with a large effect size in positive and total symptoms, global clinical severity and real-world functioning, and a moderate effect size on negative symptoms. Improvement in total symptoms (*p* < 0.001) and in global clinical severity (*p* = 0.007) emerged as individual predictors of functional improvement. These findings, although preliminary, suggest that an integrated, evidence-based treatment program is feasible and effective in a real-world rehabilitation context, and that similar interventions should be further implemented in everyday clinical practice.

## 1. Introduction

### 1.1. Background

Schizophrenia represents one of the most debilitating mental health disorders, and is often associated with significant deficits in neuro-cognitive [1,2] and socio-cognitive performance [3], as well as with marked impairments in functional capacity [4] and social skills [5].

These deficits are likely to have an impact on various everyday functional skills, such as initiating and maintaining social relationships, entering and maintaining paid jobs, living independently in the community as well as managing selfcare, healthcare and basic financial resources, resulting in a significant impairment in real-world functional outcomes [6,7].

The trajectory of schizophrenia is quite heterogeneous, with a clinical course characterized by alternating remission and relapses in about 75% of cases, according to recent meta-analyses [8,9]. Functional recovery in patients diagnosed with schizophrenia appears to be predicted by better premorbid adjustment, lower duration of untreated psychosis, better cognitive performance, concurrent remission of both positive and negative symptoms and, to a minor degree, by female gender, a better education and work history and less severe symptoms [10]; however, treatments that have been proven effective in reducing the severity of schizophrenia symptoms do not consistently show a parallel improvement in patients’ real-world functioning [11,12]. Impairment in real-world community functioning is a complex phenomenon, to which multiple, interacting factors contribute in different proportions: Health status, physical fitness, employment situation, financial disadvantage, resilience and internalized stigma, to mention a few, have a relevant impact; indeed, cognitive performance, social cognition, impaired functional capacity, availability of treatment and clinical symptoms, including depressive ones, appear to have a major role in determining everyday disability in patients diagnosed with schizophrenia [13,14].

From this perspective, functional outcome is becoming a priority target for therapeutic interventions in schizophrenia, which should be measured as a crucial part of treatment response; moreover, an increasing number of studies is showing that an integrated approach involving pharmacotherapy and psychosocial interventions is necessary to achieve the goal of functional recovery [15].

The efficacy and the effectiveness of different rehabilitation interventions in improving functional outcomes and promoting recovery has been solidly proven and is the focus of a growing body of recent scientific literature [16]. Psychosocial and rehabilitative interventions with a robust amount of evidence attesting their efficacy include structured psychoeducation [17,18], cognitive behavioral therapy [19,20], cognitive remediation [21] and social skills training [22], as well as to a lesser extent vocational interventions [23], supported housing [24] and physical exercise [25,26]; these interventions appear to be more effective and have a greater impact on functional outcomes when combined into integrated, multicomponent programs. However, implementing evidence-based, person-centered, recovery-oriented psychosocial interventions in everyday clinical practice and delivering effective psychiatric rehabilitation to patients in the real-world care settings are objectives that still have to be achieved in a large part of high-income countries, and currently represent a challenge for mental health services on a global level [27,28,29].

Among the different evidence-based interventions that can be offered to patients in real-world rehabilitation settings, cognitive remediation has shown consistent effectiveness, not only on cognitive and clinical outcomes, but also in improving psychosocial functioning [21,30,31,32,33,34].

Cognitive remediation therapy is an intervention based on behavioral training that aims at improving cognitive processes with the fundamental goals of durability and generalization, based on the strong connection linking cognition and functioning [21]. Various techniques of cognitive remediation, based on these same principles and with similar efficacy, have been developed and validated in recent years, and computer-assisted cognitive remediation (CACR) is one that can be practically implemented in rehabilitative settings [35,36,37]. However, cognitive remediation appears to have a greater positive impact on global functioning when it is combined with other evidence-based psychosocial interventions [38,39,40].

Social skills training (SST) in particular has shown consistent positive results in patients with schizophrenia [22] and can be considered an evidence-based treatment option [41,42]. SST programs, although with different durations and schedules, are based on social learning theory and are designed with the objective of improving social capacity and teaching new skills, and include goal setting, role modeling, behavioral rehearsal, positive reinforcement, corrective feedback and homework assignments. As in cognitive remediation, generalization to the community is an essential objective of SST [43].

Treatment programs combining cognitive remediation therapy and SST for patients with schizophrenia have shown promising results [44,45]. In particular, an interesting and recent randomized controlled study investigating a treatment protocol composed of an extended CACR program and group SST delivered in parallel has shown a positive effect of the intervention on cognitive performance in the working memory and in patients’ quality of life; however, no assessment of symptoms severity or other clinical measures and no assessment of the patients’ real-world functioning was carried out [46]. A recent randomized controlled trial conducted on a large sample of patients with severe mental illness has compared the effects of clinical and cognitive measures on vocational and real-world working outcomes after an 18-month follow-up of three different programs: vocational support integrated with CACR and SST, vocational support alone and vocational rehabilitation. Patients receiving vocational support integrated with CACR and SST had a higher number of hours dedicated to working or studying, compared to those receiving vocational support alone and vocational rehabilitation; however, no difference was observed between integrated and non-integrated vocational support on vocational outcomes, and no difference was observed between the three groups in clinical and, unexpectedly, cognitive measures [47].

As much as the positive efficacy results of this type of treatment program are noteworthy, the implementation of a complex treatment protocol composed by two different psychosocial interventions in usual, everyday rehabilitation settings and its effectiveness of real-world functional outcomes still needs further investigation. A recent multicentric study conducted on a large sample of community-dwelling patients has shown that the rate of patients diagnosed with schizophrenia following evidence-based psychosocial intervention, despite indications, is low (3.8% before initial evaluation, 35% at 1 year follow-up), but that the different evidence-based interventions, including psychoeducation, cognitive behavior therapy, cognitive remediation and SST, have a good real-world effectiveness in cognitive and clinical measures and in global functioning, which is also increased when more than one treatment is combined into an integrated approach. However, no evaluation of the specific intervention combinations was performed [48].

Moreover, not all patients equally benefit from SST [49] or from cognitive remediation treatment, and studies investigating factors that may influence responses to cognitive remediation have yielded no conclusive result on reliable moderators or on predictors of treatment effectiveness, both demographical and clinical, and this topic currently remains an object of scientific debate [50,51,52,53].

### 1.2. Aims of the Study

The aim of the present pilot study was to assess the feasibility and the effectiveness, in a real-world care setting, of a practical, integrated rehabilitation program, including pharmacological treatment, CACR and SST, for patients diagnosed with schizophrenia.

Improvement in real-world functional outcomes was considered the primary effectiveness measure, while improvement in schizophrenia symptoms severity was considered a secondary outcome. To evaluate if any clinical or demographical factor could have an impact on treatment effectiveness, predictors of improvement in real-world functioning were also assessed.

## 2. Materials and Methods

### 2.1. Study Design and Subjects

All patients accessing the inpatient rehabilitation center at the Department of Mental Health and Addictions of the Spedali Civili Hospital in Brescia from January 2018 to January 2019 with a diagnosis of schizophrenia were consecutively recruited in this prospective pilot study.

Inclusion and exclusion criteria for the study coincided with those of admission in the rehabilitation center, so no further selection of patients among those treated in the center was done. Inclusion criteria were (a) clinical diagnosis of schizophrenia or schizoaffective disorder according to DSM-5 criteria [54], confirmed by expert clinicians through clinical interviews and chart reviews; (b) aged between 18 and 60 years; and (c) clinical stability, defined as not requiring hospitalization or any major change in pharmacological treatment during the previous three months.

Exclusion criteria were (a) diagnosis of moderate or severe intellectual disability; (b) diagnosis of comorbid neurocognitive disorder or neurological diseases, including epilepsy; and (c) presence of a serious or unstable medical condition.

Patients were informed about the study and were invited to participate at admission in the rehabilitation center. Written consent to participate was provided through a signed form. The study was carried out in accordance with the Code of Ethics of the World Medical Association and the Declaration of Helsinki. The protocol was approved by the Ethical Committee of Brescia (Project Identification Code NP 2902). All precautions were taken for the management of sensitive data, and participants were not given monetary compensation for their involvement in the study.

### 2.2. Measures

All patients were assessed with standardized clinical and functional measures at admission and at discharge from the rehabilitation center. All the assessments were conducted by expert raters who were independent from those involved in the standard care or in administering the CACR and the SST to the patients.

For the assessment of real-world functioning, the Global Assessment of Functioning (GAF) was used. The GAF scale is the measure recommended by the DSM-IV-TR [55] for assessing social, occupational and psychological functioning. It is a frequently used, simple and comprehensive measure based on a clinician’s evaluation of the patient’s level of functioning and has been found to be a reliable and valid tool [56]. Higher scores on the scale report better overall functioning, ranging from 0 (inadequate information) to 100 (superior functioning).

Symptoms severity was assessed with the Positive and Negative Syndrome Scale (PANSS), a semi-structured interview including three subscales, namely positive symptoms, negative symptoms and general psychopathology showing good consistency, validity and reliability [57]. Each of the 30 items is accompanied by a specific definition as well as detailed anchoring criteria for all seven rating points, ranging from “absent” (1) to “severe” (7).

To assess global illness severity, the Clinical Global Impression—Severity scale (CGI-S) was used. The CGI was developed for use in clinical trials sponsored by the National Institute of Mental Health to provide a brief, stand-alone assessment of the clinician’s view of the patient’s global clinical state prior to and after initiating a study medication [58]. The CGI provides an overall summary measure that takes into account all available information, including a knowledge of the patient’s history, psychosocial circumstances, symptoms, behavior and the impact of the symptoms on the patient’s functioning. The CGI-S comprises a one-item measure evaluating the severity of the psychopathology, ranging from 1 to 7, with higher scores reporting a more severe clinical condition.

### 2.3. Treatment Program

The rehabilitation center of the Spedali Civili Hospital in Brescia is an open inpatient facility funded by the National Health System, providing psychiatric and psychosocial treatment. The rehabilitation regimen is provided to all patients and is representative of the usual setting and modalities of care of Italian psychiatric rehabilitation centers. Patients are usually referred by their treating psychiatrist before admission, and the length of the stay in the center varies from 6 to 12 months.

The rehabilitation treatment program was composed by CACR followed by SST, both provided in addition to the standard clinical practice of the rehabilitation center, which includes case management and group resocialization and leisure activities. CACR took place first in order to strengthen the cognitive performance of patients and enhance the effectiveness of the SST program, which each patient started at the conclusion of cognitive remediation. The treatment program was considered completed if a patient participated in at least 80% of the sessions of both CACR and SST.

A stable pharmacological treatment was considered necessary for the inclusion criteria of clinical stability and for admission in the rehabilitation center and in the study. All patients were maintained on their pharmacological treatment for the whole duration of the rehabilitation period and for the treatment program. No major pharmacological treatment changes, such as antipsychotic switch, were allowed; however, minor regimen or dosage modifications could take place as needed. Concomitant treatment with non-antipsychotic drugs, such as benzodiazepines, was allowed. The nursing staff of the rehabilitation center was tasked with administering the treatment to the patients during their stay, ensuring an adequate treatment adherence. Doses of antipsychotics taken were calculated as chlorpromazine equivalents according to the conversion proposed by Gardner [59].

### 2.4. Cognitive Remediation

The CACR used the Cogpack (Marker Software^®^) program. The program includes a series of cognitive exercises that can be divided into domain-specific exercises, targeting the cognitive areas that are known to be impaired in patients with schizophrenia (verbal memory, verbal fluency, psychomotor speed and coordination, executive functions, working memory and attention), and non-domain-specific exercises that tackle more than one domain at once and engage in culture, language and calculation skills. Most of the exercises can be adapted for the single patients, with the software automatically setting the difficulty level on the basis of the patient’s performance during the session. The software also records the performance of each patient for every session, with the possibility of providing feedback on the session and on the global progress. The CACR was administered individually three times a week, in 45-min sessions, for a total of 6 weeks, with a flexible schedule. Missed sessions were individually rescheduled.

### 2.5. Social Skills Training

Group SST followed a manualized approach [60], with a curriculum focused on basic communication skills and conversation, assertiveness and friendship skills [43], preceded by an individualized introductory training session and with day-to-day supervision providing positive reinforcement and corrective feedback. SST was provided twice a week in 45-min sessions for a total of 8 weeks.

### 2.6. Statistical Analyses

Comparison between dropouts and completers on demographical and baseline clinical variables was performed with an independent-samples *t*-test for continuous variables and with Pearson’s *χ*^2^ test for dichotomous variables. Comparison between the baseline and discharge values in functional and clinical measures was performed with a paired-samples *t*-test. Change from the baseline scores was also calculated as the mean difference for each measure. The distribution of scores for each variable was inspected for normality and for homogeneity of variance. In no case was there evidence that variables violated the assumptions underlying the use of parametric statistical procedures. To evaluate the effect size of the treatment program on the functional and clinical measures, Cohen’s *d* was calculated for each variable, taking into account the baseline and discharge scores as well as the standard deviations. Discharge and baseline values were subtracted and divided by pooled standard deviation. Result were reported as absolute values, with positive scores reflecting a positive treatment effect. A *d* of 0.2 corresponds to a small effect size, a *d* of 0.5 corresponds to a moderate effect size and a *d* of 0.8 or higher corresponds to a large effect size [61].

For the identification of predictors of effectiveness, the change in the GAF scores between baseline and discharge was considered the dependent variable. Potential predictors (clinical and demographic characteristics, baseline scores and change in clinical measures) were included in a multiple regression analysis if they were found to be significant in univariate exploratory analyses, performed by correlating continuous variables with the GAF change and using the independent-samples *t*-test for dichotomous variables. Multiple linear regressions followed a stepwise procedure. Collinearity was considered significant if the variance inflation factor (*VIF*) exceeded 4.0.

Statistical analyses were performed using SPSS 15.0 software (SPSS Inc., Chicago, IL, USA), and *p*-values < 0.05 (2 tailed) were considered significant. For comparison of the baseline and discharge values in the clinical and functional measures, the level of significance was adjusted taking into account correction for multiple comparisons according to the Bonferroni formula and set to *p*-value < 0.01, in order to exclude false positives (or type I errors). For the multiple linear regression, an ancillary analysis was also performed, including predictors that achieved a more inclusive level of significance, set as *p*-value < 0.25. Cohen’s *d* values and the confidence intervals were calculated with the support of a dedicated online platform [62].

## 3. Results

### 3.1. Sample Characteristics

A total of 79 subjects were admitted to the rehabilitation center during the enrolment period and provided consent to participate in the study. Of these, 72 completed the treatment program and were included in the analyses. Comparing dropouts and completers, no significant difference in any demographical or clinical variable nor in the functional or clinical measures emerged. The final sample was characterized by the presence of 34.7% (*n* = 25) female subjects and 65.3% (*n* = 47) male subjects. The mean age of the sample was 39.08 (SD ± 11.98) years.

All the demographical and clinical characteristics of the sample are reported in Table 1.

### 3.2. Treatment Effectiveness on Functional and Clinical Outcomes

Baseline, discharge and mean difference values of the functional and clinical measures are reported in Table 2.

A significant improvement (*p* < 0.001) was observed in real-world functioning, as measured by the GAF score, with a large effect size (*d* = 1.273, 95% CI: 0.915–1.631).

A significant improvement (*p* < 0.001) was also observed in symptoms severity, as measured by the PANSS score, with large effect sizes for positive symptoms and total symptoms severity (*d* = 1.269, 95% CI: 0.911–1.627 and *d* = 1.350, 95% CI: 0.988–1.712, respectively) and a moderate effect size for negative symptoms severity (*d* = 0.563, 95% CI: 0.230–0.896).

Global clinical severity, as measured by the CGI-S score, also improved significantly (*p* < 0.001) with a large effect size (*d* = 0.849, 95% CI: 0.508–1.190).

Results of the analyses comparing the baseline and discharge values in functional and clinical outcomes are reported in Table 3.

### 3.3. Predictors of Functional Improvement

Univariate analyses, exploring potential predictors of functional improvement, showed that higher age of onset (*p* = 0.042), greater improvement in global clinical severity (*p* < 0.001) and greater improvement in total, positive and negative symptoms severity (*p* < 0.001, *p* = 0.004 and *p* = 0.026, respectively) were correlated with a greater improvement in functional outcomes.

Results of the univariate analyses are reported in Table 4.

As for the multivariate linear regression analysis, improvement in total symptoms severity (*p* < 0.001) and improvement in global clinical severity (*p* = 0.007) emerged as individual predictors of functional outcomes improvement. No significant collinearity emerged between the predictors. Setting the level of significance for inclusion in the model at *p* < 0.25 as ancillary analysis, the model did not change.

Results of the multivariate linear regression are reported in Table 5.

## 4. Discussion

The results of the present study show that an integrated treatment protocol, including stable pharmacological treatment, CACR and SST, for patients with schizophrenia is feasible in a real-world clinical rehabilitation setting, and has a positive impact on functional and clinical outcomes.

The relatively low attrition rate (8.89%), although only as an indirect measure, suggests a good tolerability and appreciation of the treatment by the patients, and is in line with results reported in studies investigating other evidence-based integrated rehabilitation programs [63,64]. The sample was composed mostly by male, young to middle-aged, moderately to markedly ill adult patients [65], with serious impairment in real-world functioning [56,66].

The treatment program showed a significant effect in all measures taken into account, with large effect sizes for real-world outcomes, positive and total symptoms severity and global clinical severity, and a moderate effect size for negative symptoms. This result is also in line with those reported in previous literature on integrated psychosocial intervention, including cognitive remediation, also considering the within-group effect of the treatment arms in controlled trials [21,33]. This result is also in line with those of a previous study reporting a positive effect on clinical and functional outcomes of both cognitive remediation and SST, as well as of other evidence-based interventions, and with the finding that the positive effect of these interventions goes beyond the specific target of the treatment (such as cognition for CACR and social skills for SST) and has good generalizability [48]. Our results partially diverge from those of another recent and important study, in which a combination of CACR, SST and structured vocational support, although having a positive effect on one real-world working outcome, did not provide an improvement on non-vocational outcomes, including clinical measures, compared to vocational support alone or vocational rehabilitation [47]. This finding, however, is discussed as unexpected by the author themselves, and the discrepancies between their results and ours could be explained by the substantial and important differences in study design; the study by Christensen et al. was a controlled randomized trial, including vocational interventions, in which the patients that were recruited and that completed the integrated intervention were highly motivated community-dwelling individuals, while the patients in our study had a more severe clinical situation, requiring structured rehabilitation in a dedicated center.

As regarding the predictors of treatment response, a higher age of onset and symptoms and clinical improvement were correlated with greater functional outcomes improvement, but only total symptoms and global clinical improvement emerged as predictors at the linear regression analysis. The correlation between age of onset and improvement in real-world outcomes is in line with results reported in the literature, as a younger age of onset has been consistently found to be related with higher hospitalization rates, poorer social and occupational functioning and poorer global outcome [67]. The role of symptoms and clinical severity reduction as predictors of functional improvement is also in line with previous findings [10,15].

This study has some notable points of strength. Although complex in its structure, the treatment protocol presented in this study can be easily implement in everyday rehabilitative clinical practice, even when minimal resources are available. Indeed, computers that can be used for CACR do not require dedicated or expensive hardware, and, with appropriate training, the rehabilitation-dedicated staff already working in most centers should be able to deliver the treatment program to the patients. This fact suggests that evidence-based, recovery-oriented rehabilitation is a viable option in everyday clinical practice that should be thoroughly implemented in the usual care of patients with schizophrenia [28,68,69].

No selection of the patients accessing the rehabilitation center took place, as all patients accessing the structure in the enrollment period were included in the study and received the intervention, so the enrolled sample should be highly representative of the population of stable patients with schizophrenia requiring a rehabilitation treatment. Moreover, the tools used to assess the functional and clinical outcomes are widely known and used in routine psychiatric practice, representing practical measures of treatment response that could also be easily implemented in rehabilitation settings.

This study, however, presents some potential limitations. The absence of a control group, controlling for the effects of pharmacological therapy, usual, non-evidence-based rehabilitation activity and non-specific socialization and computer-interaction effects represents a major limitation to the evaluation of the efficacy of the present treatment program. However, this type of evaluation would be beyond the aims of the present study, which was focused more on assessing real-world feasibility and effectiveness of the integrated treatment program. Clinical and functional measures were not assessed at fixed time points, but only at admission and at discharge, leading to a possible confounding effect on functional improvement of a longer usual rehabilitation period; however, this specific aspect does not seem to be a problematic limitation, as the length of the stay in the rehabilitation center was not correlated with functional improvement (*p* = 0.697, see Table 4). No systematic assessment of the patients’ cognitive performance was taken into account, and therefore no evaluation of the treatment effect on cognitive performance, or of cognitive performance improvement on real-world outcomes, could be carried forward. However, the positive effect of cognitive remediation therapy on cognitive performance has already been thoroughly assessed in previous literature [21]; moreover, the specific training required to administer test batteries evaluating cognitive performance would have not been representative of that available in most usual, day-to-day rehabilitation contexts. No specific evaluation of social skills performance or of social functioning was also conducted. This assessment could have been interesting, in particular given the nature of the integrated treatment program; however, as for an extensive assessment of cognitive abilities, implementing an evaluation of social skills performance, which can be considered quite time-consuming and requires specific training, would have not been representative of the observed treatment setting.

Future studies should be focused on confirming these preliminary results and on evaluating the efficacy of treatment programs involving CACR and SST on real-world outcomes with a well-designed randomized controlled trial, comparing the treatment program group with an adequately designed control group, and evaluating the durability of the observed improvements, which represents one of the main objectives of both cognitive remediation and SST. Moreover, a more comprehensive evaluation of the patients’ social performance and of the patients’ real-world outcomes, using more comprehensive assessment tools, would allow to take into account the specific effect of the treatment on the different and separate areas of patient functioning.

## 5. Conclusions

Overall, the findings of the present pilot study, although preliminary, show that an integrated, evidence-based treatment program for patients with schizophrenia, consisting of stable pharmacological treatment, CACR and SST, has good feasibility and effectiveness in a real-world rehabilitation context, and suggest that similar interventions should be further implemented in everyday clinical practice.

## Figures and Tables

**Table 1 ijerph-17-03352-t001:** Characteristics of the sample.

Variable	*n*, Mean ± SD
N	72
M(%):F(%)	47(65.3):25(34.7)
Age (Years)	39.08 ± 11.98
Education (Years of Education)	10.14 ± 3.19
Age of onset (Years)	24.84 ± 7.22
Pharmacological treatment dose (CPZ eq., mg)	1040.87 ± 459.15
Length of stay in the center (Months)	7.18 ± 2.43

CPZ eq.: Chlorpromazine equivalents.

**Table 2 ijerph-17-03352-t002:** Baseline and discharge values of the functional and clinical measures.

Measure	Baseline ± SD	Discharge ± SD	Mean Difference ± SD
GAF (Functional outcomes)	32.92 ± 9.40	44.50 ± 8.78	−11.58 ± 6.62
PANSS Positive (Positive symptoms severity)	19.44 ± 4.92	13.36 ± 4.66	6.08 ± 3.48
PANSS Negative (Negative symptoms severity)	26.24 ± 4.86	23.64 ± 4.36	2.60 ± 2.75
PANSS Total (Total symptoms severity)	87.79 ± 12.61	71.28 ± 11.83	16.51 ± 8.05
CGI-S (Global clinical severity)	5.17 ± 0.63	4.47 ± 0.981	0.69 ± 0.55

GAF: Global Assessment of Functioning; PANSS: Positive and Negative Syndrome Scale; CGI-S: Clinical Global Impression—Severity.

**Table 3 ijerph-17-03352-t003:** Comparison between the baseline and discharge values of the functional and clinical measures.

Measure	*t*	*p*-Value	Cohen’s *d* (95% CI)
GAF (Functional outcomes)	−14.857	<0.001	1.273 (0.915–1.631)
PANSS Positive (Positive symptoms severity)	14.853	<0.001	1.269 (0.911–1.627)
PANSS Negative (Negative symptoms severity)	8.011	<0.001	0.563 (0.230–0.896)
PANSS Total (Total symptoms severity)	29.213	<0.001	1.350 (0.988–1.712)
CGI-S (Global clinical severity)	10.764	<0.001	0.894 (0.508–1.190)

GAF: Global Assessment of Functioning; PANSS: Positive and Negative Syndrome Scale; CGI-S: Clinical Global Impression—Severity. Cohen’s *d*s are reported as absolute values. Positive values reflect a positive treatment effect.

**Table 4 ijerph-17-03352-t004:** Univariate analyses exploring potential predictors of functional improvement.

Variable	*t*	*p*-Value
Sex	1.142	0.258
	**Pearsons’ *r***	***p*-Value**
Age (Years)	0.046	0.700
Education (Years of Education)	0.050	0.680
Age of Onset (Years)	−0.249	0.042 *
Pharmacological treatment dose (CPZ eq., mg)	−0.019	0.874
Length of stay in the center (Months)	−0.047	0.697
Baseline PANSS Positive (Positive symptoms severity)	−0.105	0.380
Baseline PANSS Negative (Negative symptoms severity)	0.014	0.907
Baseline PANSS Total (Total symptoms severity)	−0.144	0.226
Baseline CGI-S (Global clinical severity)	−0.047	0.692
PANSS Positive Change (Positive symptoms improvement)	−0.337	0.004 **
PANSS Negative Change (Negative symptoms improvement)	−0.263	0.026 *
PANSS Total Change (Total symptoms improvement)	−0.510	<0.001 **
CGI-S Change (Global clinical improvement)	−0.482	<0.001 **

* *p* < 0.05, ** *p* < 0.01; CPZ eq.: Chlorpromazine equivalents; PANSS: Positive and Negative Syndrome Scale. CGI-S: Clinical Global Impression—Severity.

**Table 5 ijerph-17-03352-t005:** Predictors of functional outcome improvement.

Individual Predictor	Standardized Beta	*t*	*p*-Value	*VIF*
PANSS Total Change (Total symptoms improvement)	−0.402	−3.699	<0.001	1.152
CGI-S Change (Global clinical improvement)	−0.304	−2.799	0.007	1.152
Model *F* = 16.737, *R*^2^ = 0.343, Adjusted *R*^2^ = 0.323, *p* < 0.001				

PANSS: Positive and Negative Syndrome Scale; CGI-S: Clinical Global Impression—Severity; *VIF*: Variance Inflation Factor.

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
