# Peer review of "Evidence-Based Integrated Intervention in Patients with Schizophrenia: A Pilot Study of Feasibility and Effectiveness in a Real-World Rehabilitation Setting"

_ijerph, 2020, doi:10.3390/ijerph17103352_

Round 1

Reviewer 1 Report

The present study assess feasibility and effectiveness on clinical and real-world outcomes of an integrated treatment protocol composed of stable pharmacological treatment, computer-assisted cognitive remediation and social skills training provided in a rehabilitation center.

The study presentation is very well-described, procedures and statistics well-done and results clearly presented. English Language and style need no major cheking. The protocol adopted is linear, although the authors list in the discussion some limitations of the study, such as the lack of a control group and the lack of a systematical assessment of patients’ cognitive performance.

However, the paper could be of interest of the Readers also in this preliminary form.   

Author Response

Reviewer 1:

The present study assess feasibility and effectiveness on clinical and real-world outcomes of an integrated treatment protocol composed of stable pharmacological treatment, computer-assisted cognitive remediation and social skills training provided in a rehabilitation center. 

The study presentation is very well-described, procedures and statistics well-done and results clearly presented. English Language and style need no major cheking. The protocol adopted is linear, although the authors list in the discussion some limitations of the study, such as the lack of a control group and the lack of a systematical assessment of patients’ cognitive performance.

However, the paper could be of interest of the Readers also in this preliminary form. 

We thank the Reviewer very much for this positive comment.

Reviewer 2 Report

There is an interesting paper addressing the effectivity of different rehabilitation interventions in patients with schizophrenia.

There are several points.

General:

The authors claim that the intervention is integrated, but it really seems that the only interventions that assess are CACR and SST. I miss the explanation about the other interventions realized or the modality of treatment used.

Abstract:

I miss the p-value for predictors.

Introduction:

I think that introduction is addressed only to the two intervention marked in the paper. It is necessary to make a more holistic introduction, indicating the effectivity of the multicomponent programs of rehabilitation and type of health attendance and relating it with the potential contribution of CACR and SST. Moreover, more evidences about community functioning in schizophrenia should be stated, defining clearly what was understand with community functioning or the real-word functioning.

I also miss some information about predictors of schizophrenia course or outcomes.

Methods:

The authors should state more accurately all the features of the treatment realized.

Why authors have not assessed social skills or social-functioning, as a measure of real-world functioning? Mainly, when an intervention in this area was done.

I think that assessment is poor, although GAF is a good tool, SOFAS could also be applied and it is also very operative to pass.

Results:

Page 5, line 192, p instead P.

Discussion & Conclusions

Discussion and conclusions should be arranged according data from integrated intervention and results from other assessment measures of real-world functioning for schizophrenia.

It seems strange that length of the stay did not correlate with the functional improvement… and the intensity of treatment?

It should to be state in the limitations the fact that authors did not evaluated social functioning.

Author Response

Reviewer 2: 

There is an interesting paper addressing the effectivity of different rehabilitation interventions in patients with schizophrenia.

There are several points.

General:

The authors claim that the intervention is integrated, but it really seems that the only interventions that assess are CACR and SST. I miss the explanation about the other interventions realized or the modality of treatment used.

In order to better explain the intervention realized, and to improve the clarity regarding the modalities of treatment used, the “Treatment program” subsection of the Methods was extensively reworked and enhanced. The intervention is defined as integrated rehabilitation as it is composed of stable pharmacological antipsychotic treatment, a cognitive remediation program and another form of psychosocial treatment, such as SST.

Abstract:

I miss the p-value for predictors.

As suggested by the Reviewer, p-values for predictors have been added to the abstract.

Introduction:

I think that introduction is addressed only to the two intervention marked in the paper. It is necessary to make a more holistic introduction, indicating the effectivity of the multicomponent programs of rehabilitation and type of health attendance and relating it with the potential contribution of CACR and SST. Moreover, more evidences about community functioning in schizophrenia should be stated, defining clearly what was understand with community functioning or the real-word functioning.

According to the request of the Reviewer, the “Background” subsection has been enhanced to provide a more holistic and comprehensive introduction, including more information and references on the effects of different evidence-based rehabilitation interventions and on the current understanding of real-world functioning in patients diagnosed with schizophrenia.

I also miss some information about predictors of schizophrenia course or outcomes.

According this comment, information regarding predictors of schizophrenia course, taking into account the outcome of recovery, have been added to the “Background” section. More references on predictors of cognitive remediation treatment have also been added.

Methods:

The authors should state more accurately all the features of the treatment realized.

The “Treatment program” subsection of the Methods was reworked and enhanced in accordance to the first comment, to the present comment and to a comment of Reviewer 3, in order to define in a more accurate way all the features of treatment realized.

Why authors have not assessed social skills or social-functioning, as a measure of real-world functioning? Mainly, when an intervention in this area was done.

We thank the Reviewer for this important question. We agree with the Reviewer that specific measures of social skills such as the Social Skills Performance Assessment (SSPA) or of social functioning such as Personal and Social Performance (PSP) or such as the Social Interactions subscale of the Specific Level Of Functioning (SLOF) could have been very interesting given the nature of the treatment program. However, these evaluations are complex and lengthy and some of them require specific training. These factors consistently limit their use in real-world, publicly funded, day-to-day rehabilitation center, which is the setting that the present study aimed to examine.

However, we have further discussed this issue in the “Discussion” section, we have added the absence of a specific measure of social skills performance and of social functioning as a limitation of the study, and added the evaluation of social functioning as a future perspective.

I think that assessment is poor, although GAF is a good tool, SOFAS could also be applied and it is also very operative to pass.

We agree with the Reviewer that a more comprehensive evaluation of real-world functioning would have been of great interest and relevance. The prospect of applying more elaborate assessment tools as a future perspective has been added to the “Discussion” section.

Results:

Page 5, line 192, p instead P.

The line has been corrected as suggested.

Discussion & Conclusions

Discussion and conclusions should be arranged according data from integrated intervention and results from other assessment measures of real-world functioning for schizophrenia.

According to this comment and to the comments of the other Reviewers, the “Discussion” and “Conclusions “ sections has been expanded and improved, providing also more a more extensive comparison with the regards of previous study findings.

It seems strange that length of the stay did not correlate with the functional improvement… and the intensity of treatment?

The length of the stay was not correlated with functional improvement, however the intensity and the total duration of both psychosocial interventions were the same for each patient. The “Treatment program” subsection has been reworked and expanded also to enhance clarity on this point following this comment.

It should to be state in the limitations the fact that authors did not evaluated social functioning.

According to the present comment and to the previous one regarding the lack of a social performance assessment, we have modified and expanded the limitations of the study.

Reviewer 3 Report

According to the authors, the aim of this work was to determine the feasibility and effectiveness on clinical and real-world outcomes of an integrated treatment protocol composed of stable pharmacological treatment, computer-assisted cognitive remediation (CACR) and social skills training provided in a rehabilitation center for patients with schizophrenia. The results showed a significant improvement in positive, negative and total symptoms, in global clinical severity and real-world functioning, with improvement in total symtoms and in global clinical severity emerging as predictors of functional improvement.

This paper is generally well-written, but concern raise regarding the thoroughness of the literature review, how this study adds to the field of study, and given these weaknesses, the discussion section. The main concern refers to the fact that it is very difficult to know what has been assessed, as the pharmacological treatment regimen could be modified during the study, and the CACR program was adapted for the single patients. Moreover, the study design does not guarantee the internal validity of the results, and some questions regarding the aim of the study remain unclear.

I have several questions and suggestions. I believe that the main issues relate to:

TITLE (page 1):

Given the design of this research, the title should reflect that this work constitutes a pilot study.

ABSTRACT (page 1):

Considering the fact that the design corresponds to a pilot study, the aim of this work (line 14), should be reconsidered; as a pilot study, its results constitute preliminary data.

INTRODUCTION (page 2):

Line 72: The authors state that “treatment programs combining cognitive remediation therapy and SST for patients with schizophrenia have shown promising results”. The literature review on this subject should be improved, as recent research (e.g., Christensen et al., 2019; Dubreucq et al., 2019; Lindenmayer et al., 2018) has provided additional evidence. On this basis, how this work adds to the field of study should be stated.

Christensen, T. N., Wallstrøm, I. G., Stenager, E., Bojesen, A. B., Gluud, C., Nordentoft, M., & Eplov, L. F. (2019). Effects of Individual Placement and Support Supplemented With Cognitive Remediation and Work-Focused Social Skills Training for People With Severe Mental Illness: A Randomized Clinical Trial. JAMA Psychiatry. https://doi.org/10.1001/jamapsychiatry.2019.2291

Dubreucq, J., Ycart, B., Gabayet, F., Perier, C. C., Hamon, A., Llorca, P. M., . . . Fond, G. (2019). Towards an improved access to psychiatric rehabilitation: Availability and effectiveness at 1-year follow-up of psychoeducation, cognitive remediation therapy, cognitive behaviour therapy and social skills training in the FondaMental advanced centers of expertise-schizophrenia (FACE-SZ) national cohort. European Archives of Psychiatry and Clinical Neuroscience, 269(5), 599-610. doi:http://dx.doi.org/10.1007/s00406-019-01001-4

Lindenmayer, J., Khan, A., McGurk, S. R., Kulsa, M. K. C., Ljuri, I., Ozog, V., . . . Parker, B. (2018). Does social cognition training augment response to computer-assisted cognitive remediation for schizophrenia? Schizophrenia Research, 201, 180-186. doi:http://dx.doi.org/10.1016/j.schres.2018.06.012

Lines 83-84: The authors state that “studies investigating factors that may influence response to cognitive remediation have yield no conclusive results on reliable moderators”. However, this study will not overcome this limitation, as it is aimed to examine predictors.

Aims of the study. Line 86: Please, see the comments for the abstract section.

MATERIALS AND METHODS (page 3)

Study design and subjects (page 3):

Further information regarding the design of the study should be provided.

Lines 98 to 102: Information regarding the treatment program should be moved to the “Treatment program” subsection.

Measures section (page 3):

Lines 121 to 122: “Change from baseline scores was also calculated as mean difference for each measure”. This sentence should be moved to the “Statistical analysis” subsection.

Treatment program section (page 4):

The main concern refers to the fact that pharmacological treatment regimen or dosage could be modified as needed, and concomitant treatment with non-antipsychotic drugs was allowed (lines 152-153). Though this has been acknowledged by the authors as a limitation of this work, his fact, together with the fact that the computer assisted cognitive remediation (CACR) program was adapted for the single patients, and the absence of a control group, considerably limits the validity of the findings.

Statistical analysis (page 4):

A more relaxed criteria for choosing predictors for the inclusion of predictors in the multiple regression analysis could be adopted (p<.25).

RESULTS (page 5):

Sample characteristics (page 5):

In Table 1, percentages for sex should be provided

In the confirmatory factor analysis section (page 4, line 148), the results of Model-1 should be explained in the text, before Table 1.

In Table 2 (page 4, line 171), measurement errors should be provided for each item for the three models.

DISCUSSION (page 5):

The discussion section should be rewritten in light of the literature review which should be added to the introduction section. More discussion with respect to the findings of previous studies is needed.

I suggest to eliminate discussion corresponding to the predictors in the univariate analysis which did not reach significance in

MINOR POINTS:                       

Page 1, line 10, “impairment” should not be written in bold.

Statistics (e.g., n, p) should be italicized.

Page 2, line 69. Please, insert a comma between “reinforcement” and “corrective”.

Page 6, line 217, “clinal” should be “clinical”.

Author Response

Reviewer 3:

According to the authors, the aim of this work was to determine the feasibility and effectiveness on clinical and real-world outcomes of an integrated treatment protocol composed of stable pharmacological treatment, computer-assisted cognitive remediation (CACR) and social skills training provided in a rehabilitation center for patients with schizophrenia. The results showed a significant improvement in positive, negative and total symptoms, in global clinical severity and real-world functioning, with improvement in total symtoms and in global clinical severity emerging as predictors of functional improvement.

This paper is generally well-written, but concern raise regarding the thoroughness of the literature review, how this study adds to the field of study, and given these weaknesses, the discussion section. The main concern refers to the fact that it is very difficult to know what has been assessed, as the pharmacological treatment regimen could be modified during the study, and the CACR program was adapted for the single patients. Moreover, the study design does not guarantee the internal validity of the results, and some questions regarding the aim of the study remain unclear.

I have several questions and suggestions. I believe that the main issues relate to:

TITLE (page 1):

Given the design of this research, the title should reflect that this work constitutes a pilot study.

According to the suggestion of the Reviewer, the title has been reworked as follows: “Evidence-based integrated intervention in patients with schizophrenia: a pilot study of feasibility and effectiveness in a real-world rehabilitation setting”

ABSTRACT (page 1):

Considering the fact that the design corresponds to a pilot study, the aim of this work (line 14), should be reconsidered; as a pilot study, its results constitute preliminary data.

As suggested by the Reviewer, the abstract has been modified, mentioning in line 14 that this represents a pilot study and in line 23 that the results of the study represent preliminary findings.

INTRODUCTION (page 2):

Line 72: The authors state that “treatment programs combining cognitive remediation therapy and SST for patients with schizophrenia have shown promising results”. The literature review on this subject should be improved, as recent research (e.g., Christensen et al., 2019; Dubreucq et al., 2019; Lindenmayer et al., 2018) has provided additional evidence. On this basis, how this work adds to the field of study should be stated.

Christensen, T. N., Wallstrøm, I. G., Stenager, E., Bojesen, A. B., Gluud, C., Nordentoft, M., & Eplov, L. F. (2019). Effects of Individual Placement and Support Supplemented With Cognitive Remediation and Work-Focused Social Skills Training for People With Severe Mental Illness: A Randomized Clinical Trial. JAMA Psychiatry. https://doi.org/10.1001/jamapsychiatry.2019.2291

Dubreucq, J., Ycart, B., Gabayet, F., Perier, C. C., Hamon, A., Llorca, P. M., . . . Fond, G. (2019). Towards an improved access to psychiatric rehabilitation: Availability and effectiveness at 1-year follow-up of psychoeducation, cognitive remediation therapy, cognitive behaviour therapy and social skills training in the FondaMental advanced centers of expertise-schizophrenia (FACE-SZ) national cohort. European Archives of Psychiatry and Clinical Neuroscience, 269(5), 599-610. doi:http://dx.doi.org/10.1007/s00406-019-01001-4

Lindenmayer, J., Khan, A., McGurk, S. R., Kulsa, M. K. C., Ljuri, I., Ozog, V., . . . Parker, B. (2018). Does social cognition training augment response to computer-assisted cognitive remediation for schizophrenia? Schizophrenia Research, 201, 180-186. doi:http://dx.doi.org/10.1016/j.schres.2018.06.012

Following the recommendations of the Reviewer, the findings of these studies have been incorporated in the “Background” subsection, and the Discussion has been modified accordingly.

Lines 83-84: The authors state that “studies investigating factors that may influence response to cognitive remediation have yield no conclusive results on reliable moderators”. However, this study will not overcome this limitation, as it is aimed to examine predictors.

We agree with the Reviewer that the mentioned statement was misleading. The sentence and the associated references have been reworked and implemented to more properly reflect the background of the present study.

Aims of the study. Line 86: Please, see the comments for the abstract section.

Following the Reviewer’s suggestion, the aims have been modified in order to mention the nature of this work as a pilot study, as has been done in the Abstract section.

MATERIALS AND METHODS (page 3)

Study design and subjects (page 3):

Further information regarding the design of the study should be provided.

Additional information has been provided regarding the study design in the “Study design and subjects” subsection.

Lines 98 to 102: Information regarding the treatment program should be moved to the “Treatment program” subsection.

Information regarding the treatment program that were reported in the “Study design and subjects” subsection have been moved to the “Treatment program” subsection according to this comment.

Measures section (page 3):

Lines 121 to 122: “Change from baseline scores was also calculated as mean difference for each measure”. This sentence should be moved to the “Statistical analysis” subsection.

As suggested, this sentence has been moved from the “Measures” to the “Statistical analysis” subsection.

Treatment program section (page 4):

The main concern refers to the fact that pharmacological treatment regimen or dosage could be modified as needed, and concomitant treatment with non-antipsychotic drugs was allowed (lines 152-153). Though this has been acknowledged by the authors as a limitation of this work, his fact, together with the fact that the computer assisted cognitive remediation (CACR) program was adapted for the single patients, and the absence of a control group, considerably limits the validity of the findings.

We agree with the Reviewers that these factors may represent a series of limitations, and, as pointed out by the Reviewer, have been acknowledged in the “Discussion” section. To further stress out the preliminary nature of the findings reported, the “Conclusion” section has been modified, highlighting this aspect.

Statistical analysis (page 4):

A more relaxed criteria for choosing predictors for the inclusion of predictors in the multiple regression analysis could be adopted (p<.25).

We thank the Reviewer for this interesting suggestion. We have performed an ancillary analysis setting the level of significance at p <0.25, which actually did not modify the results of the model. This has been reported both in the Methods and in the Results section.

RESULTS (page 5):

Sample characteristics (page 5):

In Table 1, percentages for sex should be provided

As suggested, percentages for sex were added to Table 1. 

In the confirmatory factor analysis section (page 4, line 148), the results of Model-1 should be explained in the text, before Table 1.

We had some problems in answering this comment as no confirmatory factor analysis is present in Page 4 line 148, and no factor analysis was performed in the study. Table 1 reports the characteristics of the sample. However, we moved the results of the multiple regression analysis model before the table presenting the results, in line with the rest of the results.

In Table 2 (page 4, line 171), measurement errors should be provided for each item for the three models.

Again, it was quite difficult to answer this comment, as Table 2 presents baseline and discharge values on functional and clinical measures, and in no part of the text is there a mention of three different models. However, statistical analyses and result tables were improved, also following the suggestions of Reviewer 4.

DISCUSSION (page 5):

The discussion section should be rewritten in light of the literature review which should be added to the introduction section. More discussion with respect to the findings of previous studies is needed.

As suggested by the Reviewer, the Discussion has been reworked and expanded, taking into account the findings of recent studies included in the “Background”, enhancing also the comparison with previous evidences.

I suggest to eliminate discussion corresponding to the predictors in the univariate analysis which did not reach significance in

We appreciate the suggestion of the Reviewer; however, Reviewer 5 pointed out that the correlation between functional improvement and age of onset is interesting, even if it did not reach significance beyond the univariate analysis, so we preferred to maintain the original version of the paragraph.

MINOR POINTS:                       

Page 1, line 10, “impairment” should not be written in bold.

Statistics (e.g., n, p) should be italicized.

Page 2, line 69. Please, insert a comma between “reinforcement” and “corrective”.

Page 6, line 217, “clinal” should be “clinical”.

We thank the Reviewer for pointing out these typographical errors, which have now been corrected.

Reviewer 4 Report

Nibbio et al recruited 72 patients with schizophrenia and evaluated the feasibility and effectiveness of an integrated intervention protocol on clinical and real-world outcomes. The result suggests that such treatment program is feasible and effective in the real-world rehab setups. The statistical analysis is mostly appropriate but needs to be improved. 

Specific comments:

  1. The authors should clarify how Cohen’s d and confidence intervals were computed. In Table 2, the mean differences are negative for GAF and positive for the rest of the measures. In Table 3, the Cohen’s ds are all positive, probability by taking the absolute values, but this should be clarified.
  2. The authors should apply multiple testing correction to results presented in Table 3. Such correction would not likely change the conclusion in the present study but is still important to consider.
  3. The authors explored predictors of GAF change. However, the regression analysis seems flawed. In Tables 4-5, PANSS positive/negative, PANSS total, CGI-S changes are considered as predictors. There could be correlations among them and therefore potentially causing the problem of multicollinearity. Multicollinearity happens when a variable in the regression is actually a combination of two other variables. From the definitions of these measures, PANSS total is a combination of PANSS positive/negatives, and CGI-S is an overall summary including all available information from PANSS measures to GAF, the dependent variable, itself.

Author Response

Reviewer 4:

Nibbio et al recruited 72 patients with schizophrenia and evaluated the feasibility and effectiveness of an integrated intervention protocol on clinical and real-world outcomes. The result suggests that such treatment program is feasible and effective in the real-world rehab setups. The statistical analysis is mostly appropriate but needs to be improved. 

Specific comments:

  • The authors should clarify how Cohen’s d and confidence intervals were computed. In Table 2, the mean differences are negative for GAF and positive for the rest of the measures. In Table 3, the Cohen’s ds are all positive, probability by taking the absolute values, but this should be clarified.

According to the recommendation of the Reviewer, the method used to compute Cohen’d values and confidence intervals was clarified in the “Statistical analysis” subsection.

Indeed, Cohen’s d were reported as absolute values, with positive values reflecting a positive treatment effect. This was specified both in the “Statistical analysis” subsection and in the legend of Table 3.

  • The authors should apply multiple testing correction to results presented in Table 3. Such correction would not likely change the conclusion in the present study but is still important to consider.

We thank the reviewer for this interesting comment. As suggested, correcting for multiple comparisons the results of Table 3 did not change the overall conclusions of the present work, but instead confirmed the presented findings. Significance of t-test was adapted taking into account correction for multiple comparison according to Bonferroni formula. This detail was added to the “Statistical analysis” subsection.

  • The authors explored predictors of GAF change. However, the regression analysis seems flawed. In Tables 4-5, PANSS positive/negative, PANSS total, CGI-S changes are considered as predictors. There could be correlations among them and therefore potentially causing the problem of multicollinearity. Multicollinearity happens when a variable in the regression is actually a combination of two other variables. From the definitions of these measures, PANSS total is a combination of PANSS positive/negatives, and CGI-S is an overall summary including all available information from PANSS measures to GAF, the dependent variable, itself.

This is another very interesting comment. We agree that PANSS total could present a problem of collinearity with PANSS positive and PANSS negative; however, following the stepwise multiple linear regression procedure only PANSS total emerged as individual predictor. PANSS total and CGI-S scores are not numerically related; however, we agree that symptoms severity may represent an important part of overall clinical severity and may therefore present a problem of collinearity. To account for this issues, analysis of collinearity was performed, with collinearity considered significant if VIF exceeded 4. This was added to the “Statistical analysis” subsection, to the results and to Table 5.

Reviewer 5 Report

This is an excellent study and there is very little for a reviewer to mention. In addition to providing evidence supporting combined use of social skills training and cognitive rehabilitation, the results contribute to the body of evidence indicating that the prospect of a positive treatment outcome increase with the age of onset.

The paper gives an excellent acknowledgement of the limitations of the study (thus reducing the length of this report), notably the absence of a control group. Nonetheless the paper constitutes a significant a valuable contribution.

The social skills training followed a manualized approach. Would a prior publication or online access of the manual be available?

An editorial review of English use would be appropriate (particularly in the Introduction’s Background section). The errors are minor and readily correctable. Making these corrections would enhance the impact of the paper.

Author Response

Reviewer 5:

This is an excellent study and there is very little for a reviewer to mention. In addition to providing evidence supporting combined use of social skills training and cognitive rehabilitation, the results contribute to the body of evidence indicating that the prospect of a positive treatment outcome increase with the age of onset.

The paper gives an excellent acknowledgement of the limitations of the study (thus reducing the length of this report), notably the absence of a control group. Nonetheless the paper constitutes a significant a valuable contribution.

We thank the Reviewer very much for this positive comment.

The social skills training followed a manualized approach. Would a prior publication or online access of the manual be available?

According to this comment, a reference to the employed manual was added to the “Social Skills Training” subsection.

An editorial review of English use would be appropriate (particularly in the Introduction’s Background section). The errors are minor and readily correctable. Making these corrections would enhance the impact of the paper.

Following this suggestion, the whole manuscript has been carefully revised in order to correct these errors, giving particular attention to the “Background” subsection.

Round 2

Reviewer 3 Report

The authors have answered all of my questions and included all my suggestions. Concern remains regarding the fact that the intervention differed from subject to to subject (which is valid both for pharmacological treatment [whose regimen could be modified during the intervention] and for the CACR program). This has adequately been acknowledged by the authors in the “Discussion” section: however, this limitation still constitutes a threat to the validity of the results.

Author Response

Reviewer 3      

The authors have answered all of my questions and included all my suggestions. Concern remains regarding the fact that the intervention differed from subject to to subject (which is valid both for pharmacological treatment [whose regimen could be modified during the intervention] and for the CACR program). This has adequately been acknowledged by the authors in the “Discussion” section: however, this limitation still constitutes a threat to the validity of the results. 

Following the Reviewer’s comment, we have specified in the “Treatment program” subsection that no major pharmacological treatment changes, such as antipsychotic switch, were allowed, and only minor regimen or dosage modifications could take place.

The CACR program had the same overall duration for all patients, and the Cogpack exercised were automatically adapted by the program, allowing better individualization and in accordance with the Cognitive Remediation principles of scaffolding and errorless Learning (Cognitive Remediation Therapy for Schizophrenia: Theory and Practice 1st Edition by Wykes, Til, Reeder, Clare 2005).

Reviewer 4 Report

The authors have responded reasonably well. I think it's now appropriate to publish. 

Author Response

Reviewer 4

The authors have responded reasonably well. I think it's now appropriate to publish.

We thank the Reviewer very much for this positive comment.
